# TeleFE: A New Tool for the Tele-Assessment of Executive Functions in Children

**Carlotta Rivella** [1], **Costanza Ruffini** [2], **Clara Bombonato** [3,4], **Agnese Capodieci** [2], **Andrea Frascari** [5], **Gian Marco Marzocchi** [6,*], **Alessandra Mingozzi** [6], **Chiara Pecini** [2,*], **Laura Traverso** [1], **Maria Carmen Usai** [1,*] and **Paola Viterbori** [1,*]

1   Department of Educational Sciences, University of Genoa, Corso Andrea Podestà 2, 16128 Genoa, Italy
2   Department of Education, Languages, Intercultures, Literatures and Psychology (FORLIPSI), University of Florence, Via di San Salvi 12, 50135 Florence, Italy
3   Department of Developmental Neuroscience, IRCCS Stella Maris Foundation, 56128 Pisa, Italy
4   Department of Neuroscience, Psychology, Drug Research and Child Health (NEUROFARBA), University of Florence, 50139 Florence, Italy
5   Anastasis Società Cooperativa Sociale, Via Amendola 12, 40121 Bologna, Italy
6   Department of Psychology, University of Milano-Bicocca, Piazza Ateneo Nuovo, 1, 20126 Milan, Italy
*   Correspondence: gianmarco.marzocchi@unimib.it (G.M.M.); chiara.pecini@unifi.it (C.P.); maria.carmen.usai@unige.it (M.C.U.); paola.viterbori@unige.it (P.V.)

**Abstract:** In recent decades, the utility of cognitive tele-assessment has increasingly been highlighted, both in adults and in children. The present study aimed to present TeleFE, a new tool for the tele-assessment of EF in children aged 6–13. TeleFE consists of a web platform including four tasks based on robust neuropsychological paradigms to evaluate inhibition, interference suppression, working memory, cognitive flexibility, and planning. It also includes questionnaires on EF for teachers and parents, to obtain information on the everyday functioning of the children. As TeleFE allows the assessment of EF both remotely and in-person, a comparison of the two modalities was conducted by administering TeleFE to 1288 Italian primary school children. A series of ANOVA was conducted, showing no significant effect of assessment modality ($p > 0.05$ for all the measures). In addition, significant differences by class emerged for all the measures ($p < 0.001$ for all the measures except $p = 0.008$ for planning). Finally, a significant sex effect emerged for inhibition ($p < 0.001$) and for the reaction times in both interference control ($p = 0.013$) and cognitive flexibility ($p < 0.001$), with boys showing a lower inhibition and faster reaction times. The implications of these results along with the indications for the choice of remote assessment are discussed.

**Keywords:** tele-assessment; children; executive functions; inhibition; cognitive flexibility; updating; planning

## 1. Introduction

Telehealth, which is the provision of health services remotely using telecommunications, has existed since the 1960s and includes distance-based mental and physical health services in the areas of assessment, prevention, and intervention. Within these practices, a tele-assessment of cognitive functions refers to a type of assessment in which examiners and participants interact through telecommunication technologies rather than being in the same place [1,2]. It is a relatively new practice of telehealth, and it was more widely used in adults than in children [3,4].

### 1.1. Tele-Assessment of Cognitive Functioning in Children

The importance of remote assessments of children has been increasingly emphasized in recent decades and has been exacerbated during the COVID-19 pandemic, which made it difficult to conduct traditional face-to-face assessments [5,6]. Beyond this, tele-assessment

provides several potential advantages. It allows us to overcome geographical, cultural, and socio-economical barriers, making psychological services more accessible. Tele-assessment is less expensive, both in terms of time and money, as families do not have to face long and expensive trips. For this reason, it is useful for children with medical conditions that make travel difficult, which also reduces the number of hospitalizations. It is less time-consuming for clinicians, as the scoring is automatic and immediate at the end of the test, reducing costs and increasing the number of patients in care [1,7]. Cognitive assessment via communication technologies can be considered an ecological tool to evaluate cognitive functioning in children, as it reflects the digitalization of many educational contexts and it increases motivation and engagement as reported by both parents and children [1,8,9].

Despite several advantages, there is still resistance to the use of remote assessment for children. One reason is that clinical and educational settings are typically based on face-to-face interactions. The absence of these direct interactions between the examiner and the child in remote assessments may have several consequences for the feasibility and acceptability of this modality. Privacy and data protection too are fundamental issues to be considered [10]. In addition, the complexity of the technical issues required by the remote assessment may require long training for the examiners and the participants [8,11]. Moreover, the most important variable to consider when choosing a remote assessment is the presence of inequalities in the population, in terms of the accessibility to technologies [11]. All these issues have not been studied systematically, and no clear conclusions have been reached concerning the validity of cognitive tele-assessment. The existing literature demonstrates the validity of the cognitive tele-assessment of children focused on an adapted tele-assessment, which involves the administration of instruments that are designed to be administered face-to-face in a computer-mediated manner [2,4]. To a lesser extent, instruments have been designed and standardized for digital administration (i.e., the *Automated Working Memory Assessment* [12] and the *Cambridge Neuropsychological Test Automated Battery* [13]), but not one with an Italian validation or remote standardization. Evidence from a recent systematic review [4] comparing remote and in-person assessment of the cognitive functions of children found no differences between the two modalities, suggesting that tele-assessment is feasible in the developmental age to evaluate several aspects such as languages, memory, and executive functions (EFs).

### 1.2. Executive Functions Assessment

EFs are a set of cognitive processes implicated in goal-directed behaviors that regulate our own thoughts and behaviors, especially in complex and new circumstances when our automated responses are not efficient [14]. EFs comprise three main basic components [14–16]: inhibition, which is the capacity to suppress inappropriate responses or behaviors and to control interferences of non-relevant stimuli; updating the working memory, which is the active manipulation of the information temporarily maintained in the memory; and shifting (or cognitive flexibility), which is the ability to change mental strategies, responses, or activities according to different rules or objectives. From these, the higher-order EFs are built, such as reasoning, problem-solving, and planning.

The assessment of the EFs faces several critical issues [17]. One of the most important problems is the so-called "impurity problem" [14,18], which includes different aspects. First, the EFs refer to a set of different but interrelated abilities that are not completely independent of each other. For this reason, EF tasks generally require multiple executive processes to be completed. For example, tasks measuring cognitive flexibility generally involve complex instructions that need to be kept in mind and activated according to some kind of rules, thus requiring working memory skills. So, even if different tasks have been proposed to measure specific EF components, there are no "pure" measures of any EF skills. Second, since the EFs operate by definition on other cognitive processes, any executive task will involve both the EFs and other cognitive processes not relevant to the target EF, producing difficulties in accurately measuring the executive processes [19]. An additional aspect deals with the differentiation of EFs during the course of development. The tripartite

structure found in adulthood appears distinguishable from middle childhood [20], and it is undifferentiated or otherwise organized earlier [21]. Thus, the same task may require dissimilar processes when executed at different ages.

Another important issue concerns the use of questionnaires to assess EF. While performance-based tasks are a direct measure of the child's cognitive skills, questionnaires are assumed to represent behavioral manifestations of children's underlying EF skills [22]. When they are used to assess EF in children, they are typically completed by parents and teachers, and, in the case of older children and adolescents, self-report scales are also available. One of the advantages of using questionnaires is the opportunity to obtain a more ecological measure of the EFs, as parents and teachers can observe children's behaviors in multiple contexts of everyday life. On the contrary, performance-based tasks measure the EFs under controlled conditions and may not be indicative of a child's typical daily use of those skills. In addition, questionnaires are easier to administer compared to performance-based measures. Despite these advantages, questionnaires and performance-based measures are poorly correlated. A meta-analysis of 20 studies showed that in only 24% of the studies, correlations between performance-based and questionnaire-based assessments of the EFs were statistically significant, and also in these cases the average correlation was low (r = 0.19) [23]. Different interpretations of this dissociation emerged from the literature. While some authors highlight the low validity of the indirect measures [24–26], others suggest that these results may reflect the existence of two partially dissociable domains of EF. According to this hypothesis, the direct measure of EF taps the cognitive dimension, while the indirect measures reflect the behavioral one [27]. Even if the debate is still open, it is reasonable to consider direct and indirect measures as complementary instruments to obtain a more complete description of the individual's functioning, which includes both the child's cognitive functioning and the parent and teacher impressions of the child's behaviors in everyday contexts.

A tele-assessment of EF allows us to reduce these difficulties as it allows us to measure different processes in different conditions and to precisely record both the accuracy and speed [12,13]. Specifically, thanks to the computerized assessment, it is possible to gather multiple indices that allow a more precise representation of the subjects' EF profile. For each task, it is possible to obtain measures of stimuli detection in terms of accuracy and speed (indices of *basic processes*) and measures of EF components: inhibition, interference control, cognitive flexibility, updating, and planning.

*1.3. Aims and Scope*

Considering the above-mentioned lack of research on the comparison between traditional and tele-assessment, the present study aims to explore the effects of assessment modality (remote vs. in-person) on different EF measures in school-age children and adolescents using a new battery for tele-assessment called the TeleFE (Tele Executive Function, Anastasis Cooperativa, Project RiDi Fisr2020). TeleFE consists of a web platform including four tasks based on robust neuropsychological paradigms to evaluate inhibition (Go/NoGo and flanker tasks), updating (N-back), cognitive flexibility (flanker task), and planning (daily planning task). It also includes a questionnaire on EF for teachers and parents to obtain information on the everyday functioning of the children. The test battery has been developed for children from class 1 to class 8 with the only exception of the planning task, which can be administered starting from class 3. Class- and sex-related differences have been investigated jointly with assessment modality (AM). The variability across classes and tasks, the internal consistency, and the association between the direct and indirect measures of EF components were also investigated.

## 2. Materials and Methods

*2.1. Participants*

An initial sample of 1406 children from 1st to 8th grade was recruited in primary and secondary schools located in different regions in the north, center, and south of Italy.

Children with neurologic disorders (i.e., stroke) or intellectual disabilities reported by parents and/or teachers were excluded from the final sample. The final sample comprises 1288 children ($M_{age}$ = 9.19 SD = 1.92, range 6–14 years, 633 females). The schools included are almost all public schools with users of middle or middle-low socioeconomic status and a percentage of students with non-Italian nationality of 20–30%, which is also shown by the percentage of children indicated by their parents as bilingual (17%).

*2.2. Procedure*

The web platform TeleFE (Anastasis Cooperativa, Project RiDi Fisr2020) was administered to each child via PC. All participants were individually assessed, either in-person (43%) or remotely (57%), by a graduated student trained in the use of the web platform TeleFE. Children were in a quiet classroom or room sitting 30–50 cm from the screen. The screen size was usually 11 inches (range 9–13); regardless of the AM used, before the beginning of the assessment, the adult assisting the children was required to do the calibration of the stimuli by matching a one Euro coin onto its image on the screen. In the case of remote evaluation, the children were connected with the administrator via a video conferencing software. They were supervised by an adult (teachers, parents, or trained graduate students) for technical issues. The adult was not actively involved in the administration of the tasks and was instructed to be quietly present in the room, out of sight of the child, so as not to interfere with the assessment. Connectivity to ethernet (Wi-Fi or hotspot) was checked before each session. The evaluation session lasts about 1 hour, including breaks if needed. The order of administration of TeleFE tasks was defined according to a Latin square procedure. Three different orders were used within each group (SLD, TD): order 1. flanker, Go/NoGo, and N-back; order 2. Go/NoGo, N-back, and flanker; order 3. N-back, flanker, and Go/NoGo. The TPQ task was administered to a sub-sample of children from classes 3 to 8. The task was administered as the fourth task for all the children.

In order to reduce the risk of erroneous behavior that could distort the results, such as a key pressed with not enough strength or an incorrect button pressed, each task includes a training session. In the first phase of the training session, the buttons that the children have to press appear on the screen. Then, a trial of the task appears, and the child is asked to press the correct button. In case of remote administration, the examinator sees on the screen if the child's response is correct or not. In case of an incorrect response, the examinator can verify if it is due to a misunderstanding of the task or to an erroneous behavior. The training session ends when the examinator is sure that the child has understood the task and the right behaviors during the task. If the child fails the training session, the task was not administered.

Parents and teachers of a sub-sample were asked to complete the QUFE to evaluate the EF of the child in everyday life ($N_{parents}$ = 600; $N_{teachers}$ = 559). The questionnaires were sent via email to teachers and parents. The email contains a link that allows filling out the questionnaire and automatically saving the answers in the TeleFE platform. Once completed, the clinician will be able to obtain automatic scoring.

*2.3. Measures*

**Go/NoGo task.** This task measures response inhibition, i.e., the child's ability to suppress automatic behaviors and to perform an alternative action required by the task [15,28]. A series of geometric figures (yellow or blue triangles or circles) are presented on the screen, and children are instructed to respond to a target stimulus and to avoid responding to non-target stimuli (Figure 1). The task consists of 4 blocks of 50 items (35 Go and 15 NoGo). In the 1st block, the Go stimuli are the yellow figures (regardless of the geometric shape) and the NoGo stimuli are the blue figures. In the 2nd block, the pattern reverses. In the 3rd block, the Go stimuli are circles (regardless of the color) and the NoGo stimuli are triangles. In the 4th block, the pattern reverses. Each stimulus is presented for 500 ms. When the child responds or passes the 500 ms, the stimulus disappears and a neutral screen (black without stimuli, ISI range: 500, 750, or 1000 ms) followed by the subsequent stimulus are presented.

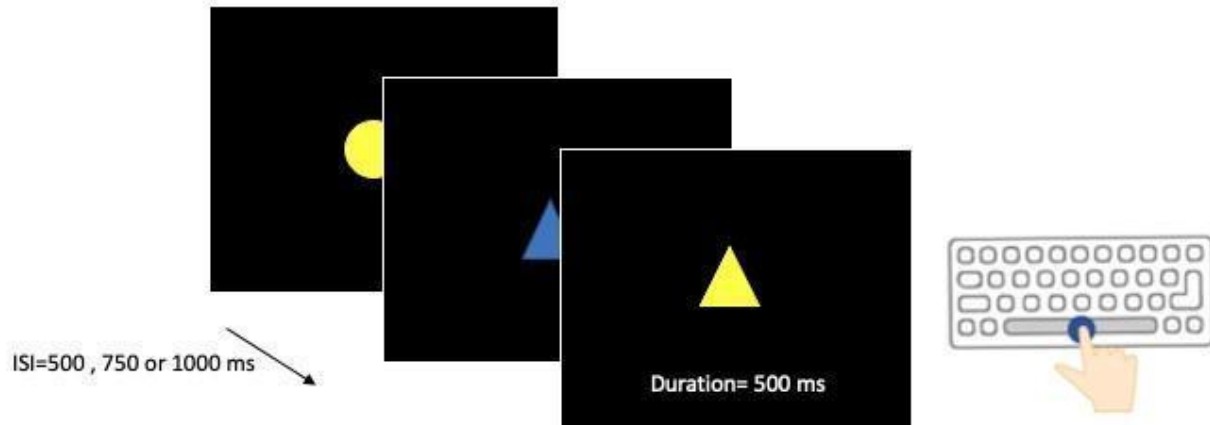

**Figure 1.** Exemplification of Go/NoGo task.

The Go/NoGo task provides the following measures: mean number of correct responses to the Go stimuli (Go CR); mean number of correct responses to the NoGo stimuli (NoGo CR); and mean number of reaction times to the Go stimuli (Go RT). NoGo CR was considered a measure of inhibition (EF component), while Go CR and Go RT represented measures of stimuli elaboration, in terms of accuracy and speed (basic processes), and they are, respectively, used as measures of task adherence and speed processing.

**Flanker task.** This task measures interference control, which is the subject's ability to ignore irrelevant information, and cognitive flexibility, which is the ability to switch between two rules according to the stimuli characteristics [29,30]. In this task, strings of five aligned arrows appear on the screen and the child is required to indicate the direction of the target arrow and to ignore the others (Figure 2). The task consists of 3 blocks. The 1st and 2nd blocks include 8 examples and 40 trials each; the 3rd block includes 64 items. In the 1st block, the arrows are blue, in the 2nd block they are orange, and in the 3rd block they are 50% blue and 50% orange. In each block, the arrows point all to the right or all to the left (congruent condition) in 50% of the trials, whereas in the other 50%, the arrow in the center points in the opposite direction to those on the sides (incongruent condition). In the 1st block (center target), the child is asked to indicate the direction of the arrow in the center by pressing as soon as possible the letter (L) on the keyboard if the arrow points to the right, and the letter (S) if it points to the left. In the 2nd block (peripheral targets), the child is asked to indicate the direction of the external arrows in the same way as the previous block. In the 3rd block (mixed rules), if the arrows are blue, the rule of the 1st block (central) must be applied; on the contrary, if the arrows are orange, a switch to the rule for the 2nd block (peripheral) is needed. In all blocks, the child sees a fixation point in the middle of the screen for about 600–1200 ms. After a white screen lasting 600 ms, the stimulus is presented for about a maximum duration of 1500 ms. A response is considered valid if it is given at least 200 ms after the stimulus is presented and before its disappearance.

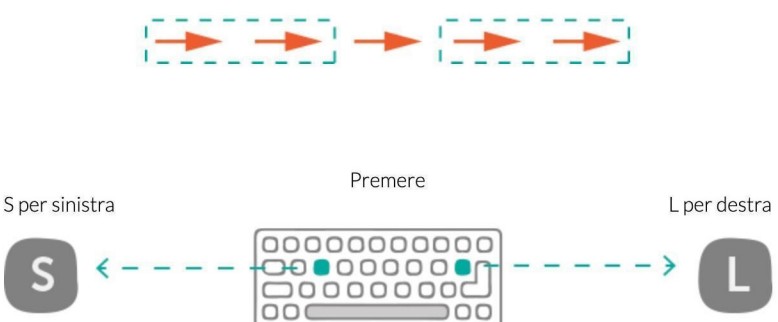

**Figure 2.** Flanker task, exemplification of the 2nd block, peripheral congruent condition.

The flanker single-rule task provides the following measures: number of correct responses in the congruent condition of the center and peripheral target blocks (congruent CR); mean of the reaction times to the correct responses in the congruent condition of the center and peripheral target blocks (congruent RT); number of correct responses in the incongruent condition of the center and peripheral target blocks (incongruent CR); and mean of the reaction times to the correct responses in the incongruent condition of the center and peripheral target blocks (incongruent RT). The flanker mixed-rule task provides the following measures: number of correct responses in the congruent condition (mixed congruent CR); mean of reaction times to the correct responses in the congruent condition (mixed congruent RT); number of correct responses in the incongruent condition (mixed incongruent CR); and mean number of reaction times to the correct responses in the incongruent condition (mixed incongruent RT). Scores in the incongruent conditions were considered measures of interference control in the single-rule task and pure measures of cognitive flexibility in the mixed-rule task (EF components). Scores in the congruent conditions were considered measures of multiple stimuli elaboration and task adherence (basic processes).

**N-back.** The N-back task is commonly used to measure updating in working memory [31,32]. The child sees a sequence of stimuli in the middle of the screen and is required to press the spacebar when the stimulus matches one of the previous stimuli (Figure 3). This task consists of 3 different conditions of increasing difficulties, which are colors, shapes, and letters. Each condition has two blocks, 1-back and 2-back, respectively, for a total of six different blocks. Stimuli (3 cm) are colors (yellow, blue, green, and red) in the first 2 blocks, shapes (triangles, circles, squares, rhombus, and pentagon) in the 3rd and 4th blocks, and letters (l, m, g, t, and b, written in both uppercase and lowercase) in the last two blocks. The child is required to respond by pressing a spacebar if the stimulus has the same color (or shape or letter) as the previous one (1-back) or as the stimulus two back (2-back). In each block there are 52 items (16 targets). Stimulus presentation time is 1500 ms with an ISI of 1000 ms.

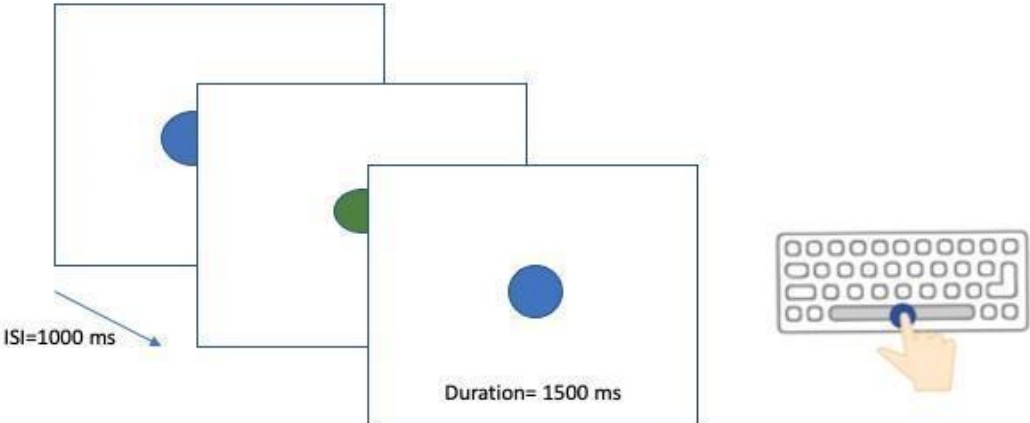

**Figure 3.** Exemplification of the 2-back task.

The N-back task provides the following measures: number of correct responses in the 1-back blocks (colors, shapes, and letters; 1-back); number of responses to the targets plus number of absences of responses to the non-targets in the 2-back blocks (colors, shapes, and letters; 2-back). The 1-back and 2-back scores were considered a measure of updating in working memory in a low and high load, respectively (EF component).

**Daily planning test (DPT).** This task, adapted from Sgaramella, Bisiacchi, and Falchero [33], measures planning ability, which is the ability to select and organize actions to reach a goal. In this task, the child is asked to organize a schedule for a hypothetical day by defining the order of the errands in order to (1) do them all, respecting logical and time constraints, and (2) follow the quickest route. A map with streets, houses, and stores and a list (in random order) of 11 activities are presented on the screen. Activities have

to be completed in different places and must consider specific constraints (e.g., "the math homework must be done at 5 p.m." or "you need to buy a bus ticket to go to the tennis class") (see Figure 4). The following activities had to be planned by the child: buy the bus ticket, have lunch at home with the family (remember, you had to buy ingredients), buy the ingredients needed to cook lunch, do Italian homework, buy a checkbook for math homework at 5 p.m., go to Lucia's home to do homework (remember, you need a new notebook), free time after doing all the homework, go to the tennis lesson at 3 p.m. by bus (remember you need to buy a ticket), pack your bag for the tennis lesson, go play with Tommy at the park after doing all the homework, and bring groceries to dad by 12 p.m. The child is shown the list of activities to plan, then asked to repeat them (recall), to estimate how long each activity takes (time estimation), and to put them in order (planning). Finally, children aged 11 years and older are asked to indicate for each action the time required to execute the activity. The planning score is represented by the percentage of errands arranged in the correct order using the map (DPT).

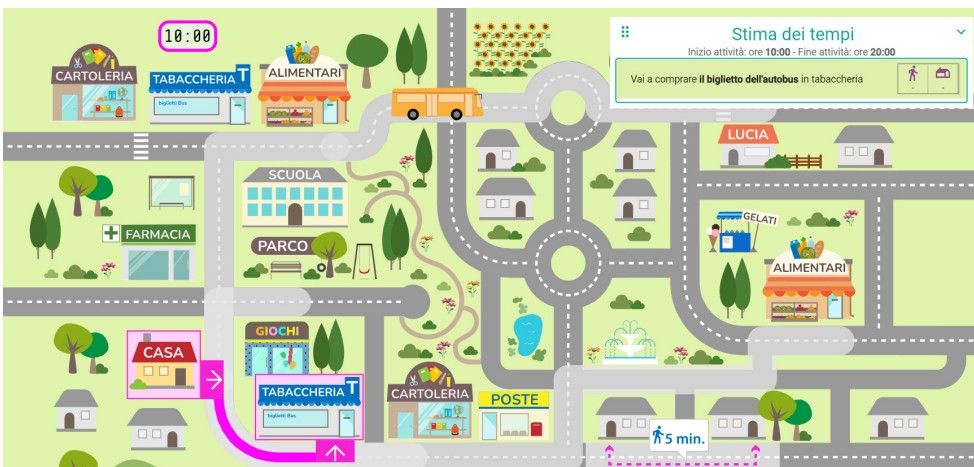

**Figure 4.** DPT map.

**Questionnaires for EF assessment (QUFE)** This questionnaire, previously standardized by Schweiger and Marzocchi [34], provides two versions for parents and teachers, respectively. It consists of 32 items concerning executive behaviors that may occur at home (or at school). For each of them, the parent (or the teacher) must make a judgment on a 5-point scale ranging from "not at all true" to "completely true"; in the parental form, the items consider 5 different components of EF: cognitive self-regulation, behavioral self-regulation, cognitive flexibility, material organization, and initiative. The teacher's form concerns cognitive and behavioral self-regulation and material organization. Finally, a total score is provided. The internal consistency is good for both the parent ($\alpha = 0.95$) and the teacher ($\alpha = 0.98$) versions.

### 2.4. Statistical Analysis

Analyses were performed on participants' characteristics and TeleFE scores using Jamovi Software 2.3.18. For each task, only children who completed the task with a total accuracy higher than 60% on the go trials, the congruent trials for the flanker task, the total trials for 1-back and 2-back, and no block with an accuracy equal to 0 in the other conditions were considered for the analysis. This allows only children who responded above the chance level to be included, which ensures that only children who understood the task and made no non-FE-related errors (i.e., moving their fingers on the keyboard) are included. Other standardized tests (i.e., NEPSY-II) [35] use a similar procedure, allowing the administration of the test only to children who pass the example trial accurately. Values above or below 3 standard deviations from the mean scores are considered outliers and were removed from the dataset.

Analyses of the normality of the distribution (skewness cut-off = 2; kurtosis cut-off = 3) were carried out on all measures. The accuracy scores of 1-back were transformed into rank scores because of the high kurtosis, and the transformed scores were used for all the analyses. A series of ANOVA tests were conducted to examine the effect of class, sex, and administration modality (AM) on each TeleFE task, both for accuracy and RTs. Repeated-measure ANOVAs were performed in the case of congruent and incongruent flanker conditions to explore the presence of a flanker effect. Corresponding post hoc (Tukey method) was also carried out to reveal the specific within- and between- factor differences. Effect size was expressed by the partial eta squared ($\eta_p^2$) values. We used the coefficient of variation (CV) SD/mean, as a measure of variability across classes of age and tasks. The internal consistency was calculated with the Cronbach alpha for each accuracy measure except for the DPT. The test–retest reliability for the EF components (except for the DPT) was calculated on a sub-sample of 54 children from classes 2 to 4 (23 male; $M_{Age}$ = 8.32; SD = 0.96). All the children were evaluated remotely; the interval between the two administrations was three months. Partial correlations with age as a control variable were used. The association between the EF components was investigated with partial correlations using age as a control variable.

## 3. Results

Preliminary analyses show that the criteria of 60% of accuracy is not reached in some cases that were excluded from the subsequent analyses: 3.4% in the Go/NoGo task, 9.1% in the first two blocks, and 13.7% in the third block of the flanker task, 6% in the 1-back, and 4.7% in the 2-back. For the flanker task, these cases belong mainly to the first three classes in the remote condition, suggesting that younger children may require more support during the remote assessment. The criterion was reached by all the participants in the DPT. The outliers in the Go/NoGo task were 2.8% considering Go CR and less than 1% considering NoGo CR and Go TR. The outliers in the flanker task range from 0.1% (mixed CR congruent) to 2.2% (CR congruent). Outliers were 0.9% in the 1-back and 0.25% in the 2-back. Finally, no outliers were removed for the DPT.

The descriptive for each task by class are shown in Table 1.

**Table 1.** Descriptive for each EF task by class.

|  | Class | N | Mean | SD | Min | Max | Skew | SE | Kurt | SE | V.C. | Post hoc ($p < 0.05$) |
|---|---|---|---|---|---|---|---|---|---|---|---|---|
| Go CR | 1 | 70 | 128.06 | 9.88 | 100 | 140 | −1.37 | 0.29 | 1.16 | 0.57 | 0.08 | <4–8 |
|  | 2 | 199 | 128.39 | 10.12 | 93 | 140 | −1.64 | 0.17 | 2.33 | 0.34 | 0.08 | <5–8 |
|  | 3 | 220 | 131.33 | 8.26 | 98 | 140 | −1.70 | 0.16 | 2.97 | 0.33 | 0.06 | <4–8 |
|  | 4 | 269 | 133.24 | 7.34 | 104 | 140 | −1.91 | 0.15 | 3.65 | 0.30 | 0.06 | >1, 3 |
|  | 5 | 183 | 134.04 | 5.82 | 110 | 140 | −1.74 | 0.18 | 3.40 | 0.36 | 0.04 | >1–3 |
|  | 6 | 88 | 135.52 | 4.36 | 121 | 140 | −1.41 | 0.26 | 1.47 | 0.51 | 0.03 | >1–3 |
|  | 7 | 91 | 135.82 | 3.57 | 123 | 140 | −1.15 | 0.25 | 1.78 | 0.50 | 0.03 | >1–3 |
|  | 8 | 89 | 136.19 | 4.70 | 118 | 140 | −1.96 | 0.26 | 4.30 | 0.51 | 0.03 | >1–3 |
| NoGo CR | 1 | 71 | 42.79 | 7.23 | 18 | 57 | −0.60 | 0.28 | 1.15 | 0.56 | 0.17 | <6–8 |
|  | 2 | 203 | 41.37 | 8.84 | 18 | 59 | −0.75 | 0.17 | 0.01 | 0.34 | 0.21 | <4–8 |
|  | 3 | 225 | 43.62 | 7.03 | 21 | 58 | −0.40 | 0.16 | −0.11 | 0.32 | 0.16 | <4–8 |
|  | 4 | 275 | 45.91 | 6.97 | 24 | 59 | −0.66 | 0.15 | 0.03 | 0.29 | 0.15 | <7–8 |
|  | 5 | 186 | 46.13 | 6.49 | 26 | 59 | −0.67 | 0.18 | 0.09 | 0.35 | 0.14 | <7–8 |
|  | 6 | 91 | 47.87 | 6.18 | 33 | 59 | −0.57 | 0.25 | −0.35 | 0.50 | 0.13 | >1–3 |
|  | 7 | 92 | 48.65 | 6.17 | 34 | 58 | −0.49 | 0.25 | −0.61 | 0.50 | 0.13 | >1–5 |
|  | 8 | 90 | 50.17 | 6.09 | 32 | 59 | −1.12 | 0.25 | 1.13 | 0.50 | 0.12 | >1–5 |
| Go RT | 1 | 72 | 2135.54 | 211.70 | 1684 | 2603 | 0.24 | 0.28 | −0.38 | 0.56 | 0.10 | >3–8 |
|  | 2 | 203 | 2015.38 | 208.53 | 1575 | 2556 | 0.38 | 0.17 | −0.32 | 0.34 | 0.10 | >4–8 |
|  | 3 | 226 | 1921.12 | 189.02 | 1344 | 2422 | 0.20 | 0.16 | 0.21 | 0.32 | 0.10 | >4–8 |
|  | 4 | 275 | 1814.62 | 187.77 | 1364 | 2320 | 0.38 | 0.15 | −0.19 | 0.29 | 0.10 | >5–8 |
|  | 5 | 186 | 1751.87 | 186.35 | 1348 | 2310 | 0.54 | 0.18 | 0.24 | 0.35 | 0.11 | >7–8 |
|  | 6 | 91 | 1684.89 | 151.58 | 1317 | 2040 | −0.06 | 0.25 | 0.06 | 0.50 | 0.09 | >8 |
|  | 7 | 91 | 1644.63 | 143.13 | 1294 | 1954 | −0.13 | 0.25 | −0.54 | 0.50 | 0.09 | <1–5 |
|  | 8 | 90 | 1575.43 | 147.55 | 1289 | 2049 | 0.66 | 0.25 | 0.49 | 0.50 | 0.09 | <1–6 |

**Table 1.** *Cont.*

| | Class | N | Mean | SD | Min | Max | Skew | SE | Kurt | SE | V.C. | Post hoc (*p* < 0.05) |
|---|---|---|---|---|---|---|---|---|---|---|---|---|
| Congruent CR | 1 | 55 | 33.84 | 4.42 | 24 | 40 | −0.61 | 0.32 | −0.59 | 0.63 | 0.13 | <3–8 |
| | 2 | 171 | 35.01 | 4.40 | 24 | 40 | −0.84 | 0.19 | −0.29 | 0.37 | 0.13 | <3–8 |
| | 3 | 210 | 36.95 | 2.88 | 24 | 40 | −1.46 | 0.17 | 2.47 | 0.33 | 0.08 | <5–8 |
| | 4 | 264 | 37.78 | 2.85 | 27 | 40 | −1.97 | 0.15 | 3.52 | 0.30 | 0.08 | <7–8 |
| | 5 | 181 | 38.62 | 1.47 | 34 | 40 | −0.96 | 0.18 | 0.10 | 0.36 | 0.04 | >1–3 |
| | 6 | 86 | 38.98 | 1.26 | 35 | 40 | −1.38 | 0.26 | 1.48 | 0.51 | 0.03 | >1–3 |
| | 7 | 91 | 39.14 | 1.23 | 34 | 40 | −2.02 | 0.25 | 4.64 | 0.50 | 0.03 | >1–4 |
| | 8 | 87 | 39.30 | 0.93 | 36 | 40 | −1.26 | 0.26 | 1.09 | 0.51 | 0.02 | >1–4 |
| Incongruent CR | 1 | 55 | 34.24 | 6.67 | 13 | 40 | −1.59 | 0.32 | 2.16 | 0.63 | 0.19 | <5–8 |
| | 2 | 171 | 28.70 | 8.90 | 9 | 40 | −0.40 | 0.19 | −1.09 | 0.37 | 0.31 | <5–8 |
| | 3 | 216 | 32.13 | 7.33 | 4 | 40 | −1.17 | 0.17 | 0.72 | 0.33 | 0.23 | <5–8 |
| | 4 | 266 | 33.48 | 7.08 | 9 | 40 | −1.37 | 0.15 | 1.01 | 0.30 | 0.21 | <6–8 |
| | 5 | 184 | 35.43 | 5.20 | 14 | 40 | −1.85 | 0.18 | 3.18 | 0.36 | 0.15 | >1–3 |
| | 6 | 87 | 36.32 | 4.34 | 19 | 40 | −2.16 | 0.26 | 5.05 | 0.51 | 0.12 | >1–4 |
| | 7 | 89 | 38.12 | 2.15 | 30 | 40 | −1.69 | 0.26 | 3.01 | 0.51 | 0.06 | >1–4 |
| | 8 | 88 | 38.09 | 2.63 | 22 | 40 | −3.15 | 0.26 | 15.40 | 0.51 | 0.07 | >1–4 |
| Congruent RT | 1 | 55 | 1842.44 | 269.50 | 1216 | 2560 | 0.10 | 0.32 | 0.12 | 0.63 | 0.15 | >3–8 |
| | 2 | 171 | 1715.68 | 242.39 | 1118 | 2431 | 0.14 | 0.19 | 0.01 | 0.37 | 0.14 | >4–8 |
| | 3 | 215 | 1619.55 | 230.58 | 1099 | 2452 | 0.48 | 0.17 | 0.57 | 0.33 | 0.14 | >4–8 |
| | 4 | 266 | 1506.05 | 240.14 | 968 | 2389 | 0.54 | 0.15 | 0.44 | 0.30 | 0.16 | >5–8 |
| | 5 | 182 | 1400.63 | 254.71 | 814 | 2500 | 0.66 | 0.18 | 1.36 | 0.36 | 0.18 | >8 |
| | 6 | 86 | 1322.67 | 216.69 | 883 | 2221 | 0.82 | 0.26 | 2.62 | 0.51 | 0.16 | >8 |
| | 7 | 89 | 1303.29 | 247.49 | 911 | 2186 | 1.07 | 0.26 | 1.33 | 0.51 | 0.19 | <1–4 |
| | 8 | 87 | 1172.03 | 202.89 | 866 | 1743 | 0.83 | 0.26 | 0.28 | 0.51 | 0.17 | <1–7 |
| Incongruent RT | 1 | 55 | 2024.55 | 288.93 | 1230 | 2717 | −0.30 | 0.32 | 0.43 | 0.63 | 0.14 | >4–8 |
| | 2 | 171 | 1897.22 | 248.53 | 1208 | 2498 | −0.28 | 0.19 | −0.11 | 0.37 | 0.13 | >4–8 |
| | 3 | 216 | 1848.92 | 275.88 | 1169 | 2673 | 0.12 | 0.17 | −0.03 | 0.33 | 0.15 | >4–8 |
| | 4 | 268 | 1705.10 | 289.86 | 1023 | 2578 | 0.39 | 0.15 | 0.02 | 0.30 | 0.17 | >5–8 |
| | 5 | 186 | 1563.60 | 292.69 | 843 | 2404 | 0.21 | 0.18 | −0.18 | 0.35 | 0.19 | >7–8 |
| | 6 | 88 | 1453.17 | 226.50 | 964 | 2078 | 0.12 | 0.26 | −0.27 | 0.51 | 0.16 | >8 |
| | 7 | 91 | 1428.35 | 259.28 | 995 | 2246 | 0.74 | 0.25 | 0.32 | 0.50 | 0.18 | >8 |
| | 8 | 90 | 1263.11 | 230.78 | 855 | 1835 | 0.65 | 0.25 | −0.21 | 0.50 | 0.18 | <1–7 |
| 1-back | 1 | 66 | 136.88 | 10.93 | 102 | 154 | −1.18 | 0.29 | 1.96 | 0.58 | 0.08 | <3–8 |
| | 2 | 179 | 138.78 | 12.50 | 94 | 155 | −1.24 | 0.18 | 1.29 | 0.36 | 0.09 | <4–8 |
| | 3 | 230 | 143.49 | 9.32 | 105 | 154 | −1.92 | 0.16 | 4.36 | 0.32 | 0.06 | <4–8 |
| | 4 | 270 | 146.67 | 7.12 | 116 | 156 | −1.62 | 0.15 | 2.76 | 0.30 | 0.05 | <7–8 |
| | 5 | 188 | 148.64 | 4.93 | 126 | 155 | −1.37 | 0.18 | 2.27 | 0.35 | 0.03 | >1–3 |
| | 6 | 87 | 148.44 | 5.83 | 126 | 156 | −1.76 | 0.26 | 3.32 | 0.51 | 0.04 | >1–3 |
| | 7 | 91 | 148.44 | 6.71 | 115 | 156 | −2.57 | 0.25 | 8.10 | 0.50 | 0.05 | >1–4 |
| | 8 | 89 | 150.45 | 3.91 | 133 | 156 | −1.65 | 0.26 | 4.01 | 0.51 | 0.03 | >1–4 |
| 2-back | 1 | 64 | 117.80 | 7.83 | 96 | 137 | −0.55 | 0.30 | 0.80 | 0.59 | 0.07 | <4–8 |
| | 2 | 170 | 118.39 | 8.58 | 94 | 140 | −0.29 | 0.19 | 0.40 | 0.37 | 0.07 | <4–8 |
| | 3 | 224 | 121.45 | 9.29 | 96 | 144 | −0.46 | 0.16 | 0.22 | 0.32 | 0.08 | <4–8 |
| | 4 | 271 | 126.98 | 8.92 | 103 | 147 | −0.22 | 0.15 | −0.24 | 0.29 | 0.07 | <7–8 |
| | 5 | 187 | 128.39 | 7.93 | 106 | 148 | −0.11 | 0.18 | −0.08 | 0.35 | 0.06 | <8 |
| | 6 | 88 | 129.26 | 8.94 | 110 | 149 | 0.01 | 0.26 | −0.63 | 0.51 | 0.07 | >1–3 |
| | 7 | 90 | 130.62 | 8.71 | 103 | 154 | −0.12 | 0.25 | 0.77 | 0.50 | 0.07 | >1–4 |
| | 8 | 89 | 133.66 | 9.15 | 110 | 153 | −0.22 | 0.26 | −0.32 | 0.51 | 0.07 | >1–5 |
| Mixed Congruent CR | 1 | 46 | 26.83 | 3.36 | 20 | 32 | −0.55 | 0.35 | −0.51 | 0.69 | 0.13 | <5–8 |
| | 2 | 153 | 27.45 | 3.57 | 19 | 32 | −0.74 | 0.20 | −0.53 | 0.39 | 0.13 | <7–8 |
| | 3 | 201 | 28.10 | 3.07 | 19 | 32 | −0.97 | 0.17 | 0.42 | 0.34 | 0.11 | <5–8 |
| | 4 | 263 | 28.51 | 3.33 | 19 | 32 | −1.07 | 0.15 | 0.33 | 0.30 | 0.12 | <5, 7–8 |
| | 5 | 181 | 29.65 | 2.21 | 23 | 32 | −1.06 | 0.18 | 0.42 | 0.36 | 0.07 | >1, 3–4 |
| | 6 | 83 | 29.60 | 2.29 | 22 | 32 | −1.32 | 0.26 | 1.57 | 0.52 | 0.08 | >1, 3 |
| | 7 | 88 | 30.17 | 2.11 | 22 | 32 | −1.67 | 0.26 | 2.90 | 0.51 | 0.07 | >1–4 |
| | 8 | 87 | 30.76 | 1.32 | 26 | 32 | −1.22 | 0.26 | 1.40 | 0.51 | 0.04 | >1–4 |
| Mixed Incongruent CR | 1 | 46 | 15.85 | 4.97 | 2 | 29 | 0.06 | 0.35 | 0.80 | 0.69 | 0.31 | <5–8 |
| | 2 | 152 | 15.89 | 4.59 | 3 | 29 | −0.03 | 0.20 | 0.06 | 0.39 | 0.29 | <4–8 |
| | 3 | 201 | 18.94 | 4.74 | 7 | 31 | 0.07 | 0.17 | −0.39 | 0.34 | 0.25 | <4–8 |
| | 4 | 262 | 20.96 | 5.20 | 7 | 32 | −0.27 | 0.15 | −0.62 | 0.30 | 0.25 | <6–8 |
| | 5 | 183 | 22.43 | 4.92 | 8 | 32 | −0.51 | 0.18 | −0.26 | 0.36 | 0.22 | <7–8 |
| | 6 | 85 | 23.27 | 4.66 | 12 | 31 | −0.45 | 0.26 | −0.58 | 0.52 | 0.20 | >1–4 |
| | 7 | 90 | 25.17 | 4.00 | 13 | 32 | −0.86 | 0.25 | 1.00 | 0.50 | 0.16 | >1–5 |
| | 8 | 88 | 25.41 | 4.56 | 13 | 31 | −0.95 | 0.26 | 0.21 | 0.51 | 0.18 | >1–5 |

**Table 1.** *Cont.*

|  | Class | N | Mean | SD | Min | Max | Skew | SE | Kurt | SE | V.C. | Post hoc ($p < 0.05$) |
|---|---|---|---|---|---|---|---|---|---|---|---|---|
| Mixed Congruent RT | 1 | 45 | 1113.02 | 147.36 | 717 | 1361 | −0.65 | 0.35 | 0.36 | 0.69 | 0.13 | >6, 8 |
|  | 2 | 153 | 1049.94 | 172.94 | 649 | 1499 | 0.22 | 0.20 | −0.40 | 0.39 | 0.16 | >8 |
|  | 3 | 201 | 1051.84 | 164.63 | 573 | 1412 | −0.24 | 0.17 | −0.21 | 0.34 | 0.16 | >6, 8 |
|  | 4 | 263 | 1030.94 | 163.86 | 559 | 1484 | −0.13 | 0.15 | −0.23 | 0.30 | 0.16 | >8 |
|  | 5 | 184 | 997.37 | 183.77 | 467 | 1495 | −0.26 | 0.18 | 0.36 | 0.36 | 0.18 | - |
|  | 6 | 85 | 969.01 | 149.37 | 602 | 1393 | 0.04 | 0.26 | 0.09 | 0.52 | 0.15 | <1, 3 |
|  | 7 | 91 | 985.71 | 158.46 | 593 | 1363 | −0.16 | 0.25 | 0.12 | 0.50 | 0.16 | - |
|  | 8 | 89 | 913.04 | 165.32 | 527 | 1395 | 0.08 | 0.26 | −0.06 | 0.51 | 0.18 | <1–4 |
| Mixed Incongruent RT | 1 | 45 | 1337.84 | 156.07 | 962 | 1692 | −0.45 | 0.35 | 0.45 | 0.69 | 0.12 | >8 |
|  | 2 | 153 | 1244.58 | 211.46 | 675 | 1688 | −0.42 | 0.20 | −0.12 | 0.39 | 0.17 | >8 |
|  | 3 | 200 | 1275.66 | 191.09 | 714 | 1644 | −0.63 | 0.17 | 0.26 | 0.34 | 0.15 | >5–8 |
|  | 4 | 262 | 1258.15 | 180.18 | 744 | 1793 | −0.11 | 0.15 | 0.08 | 0.30 | 0.14 | >6–8 |
|  | 5 | 184 | 1201.87 | 165.32 | 767 | 1597 | −0.13 | 0.18 | −0.32 | 0.36 | 0.14 | >8 |
|  | 6 | 85 | 1179.25 | 149.66 | 773 | 1513 | −0.33 | 0.26 | 0.21 | 0.52 | 0.13 | <3–4 |
|  | 7 | 91 | 1179.13 | 165.16 | 710 | 1547 | −0.23 | 0.25 | −0.20 | 0.50 | 0.14 | <3–4 |
|  | 8 | 89 | 1089.13 | 175.09 | 575 | 1497 | −0.30 | 0.26 | 0.52 | 0.51 | 0.16 | <1–5 |
| DPT | 3 | 28 | 79.43 | 9.75 | 64 | 91 | −0.32 | 0.44 | −1.13 | 0.86 | 0.12 | <4, 6–8 |
|  | 4 | 49 | 88.61 | 9.85 | 55 | 100 | −1.13 | 0.34 | 1.93 | 0.67 | 0.11 | >3 |
|  | 5 | 29 | 88.52 | 6.76 | 73 | 100 | −0.03 | 0.43 | −0.25 | 0.85 | 0.08 | - |
|  | 6 | 30 | 91.00 | 9.15 | 73 | 100 | −0.84 | 0.43 | −0.25 | 0.83 | 0.10 | >3 |
|  | 7 | 24 | 90.25 | 9.16 | 73 | 100 | −0.63 | 0.47 | −0.58 | 0.92 | 0.10 | >3 |
|  | 8 | 18 | 92.00 | 6.83 | 82 | 100 | −0.19 | 0.54 | −1.12 | 1.04 | 0.07 | >3 |

### 3.1. Basic Processes: Go/NoGo Task

The ANOVA by class, sex, and assessment modality for the Go CR shows significant differences by class ($F_{7,1177} = 10.53$, $p < 0.001$, $\eta^2_p = 0.06$). Post hoc tests reveal that the scores in class 1 and 3 differ significantly from classes 4 and above; the scores in class 2 are lower than the scores in classes 5 and above; and the scores in class 4 are lower than the scores in class 8 (all ps < 0.05; Table 1; Figure S1a). There are no significant differences from class 5 to 8. There are no sex or AM effects, not even in interaction with each other. The variability is lower than 10% for all classes (CVs range 0.03–0.08, Table 1).

The ANOVA for the Go TR shows significant differences by class ($F_{7,1202} = 71.67$, $p < 0.001$, $\eta^2_p = 0.29$) and sex ($F_{1,1202} = 10.37$, $p = 0.001$, $\eta^2_p = 0.01$). Post hoc tests reveal that the RTs in class 1 differ significantly from classes 3 and above; the RTs in class 2 and 3 are slower than the RTs in classes 4 and above; the RTs in class 4 are slower than those of the other classes; the RTs in class 5 differ from those in class 7 and 8; and the scores in class 6 differ from class 8 (all ps < 0.05). The RTs in class 7 do not differ from those in class 8 (Table 1). In addition, females show higher TR than boys, regardless of class ($M_{diff} = 47.17$, $p = 0.001$), and there are neither AM nor interaction effects. The variability is around 10% for all classes (CVs range 0.09–0.11, Table 1; Figure S1c).

### 3.2. Response Inhibition: Go/NoGo Task

The ANOVA for the NoGo CR shows significant differences by class ($F_{7,1201} = 14.47$, $p < 0.001$ $\eta^2_p = 0.08$) and sex ($F (1, 1201) = 23.98$, $p < 0.001$, $\eta^2_p = 0.02$). Post hoc tests reveal that the scores in class 1 differ significantly from classes 6 and above; the scores in class 2 and 3 are lower than the scores in classes 4 and above; the scores in class 4 and 5 are lower than the scores in class 7 and 8 (all ps < 0.05; Table 1; Figure S1b); and there are no significant differences for the classes 6 to 8. In addition, females show higher scores than boys, regardless of the class ($M_{diff} = 2.75$, $p < 0.001$), and there are neither AM nor interaction effects. The variability ranges between 12% and 21% (CVs range 0.12–0.21, Table 1).

### 3.3. Interference Suppression: Flanker Task

The repeated-measures ANOVA on accuracy (congruent CR and incongruent CR) by class, sex, and assessment modality reveals the significant main effects of the congru-

ent/incongruent condition ($F_{1,1104} = 116.76$, $p < 0.001$, $\eta^2_p = 0.10$) and class ($F_{7,1104} = 25.20$, $p < 0.001$, $\eta^2_p = 0.14$). The interaction between condition and class was significant ($F_{7,1104} = 5.91$, $p < 0.001$, $\eta^2_p = 0.04$), which demonstrates that the accuracy scores show a flanker effect from the classes 2 to 5 (all ps < 0.05) but not for the other classes. Accuracy on the congruent trials differs among the classes: class 1 and 2 differ from classes 3 and above; class 3 and 4 differ from the upper classes, while there are no significant differences from classes 5 to 8. In the case of the incongruent trials, the first four classes differ from the classes above and there are no other significant effects (Table 1; Figure S2a). The interactions class X AM ($F_{1,1104} = 3.67$, $p = 0.015$, $\eta^2_p = 0.01$), and condition X class X AM ($F_{7,1104} = 5.72$, $p < 0.001$, $\eta^2_p = 0.04$) were also significant. Post hoc analyses show that there is no difference between an assessment in-person and a remote assessment in congruent and incongruent trials for each class in the correspondent values. For example, the accuracy for the remote vs. in-person in congruent (or incongruent) trials in class 1 and so on, except for the incongruent trials in class 4, showed that children evaluated remotely were less accurate than children evaluated in-person (32.06 vs. 35.35, $p < 0.05$). In-person, there are the following significant differences between the classes on the congruent scores: 1 < 4–8; 2 < 3–8; 3 < 5; and 7–8. Remotely, there are the following significant differences between the classes on the congruent scores: 1–2 < 3–8 and 3 < 6–8. In-person there are the following significant differences between the classes on the incongruent scores: 1 < 4–8 and 3 < 7–8. Remotely, there are the following significant differences between the classes on the incongruent scores: 1 > 2; 2 < 3–8; 3 < 5–8; and 4 < 6–8 (Table S1). The variability ranges between 0.2 and 0.13, and between 0.06 and 0.31 for the congruent and incongruent correct responses, respectively, showed a decreasing trend with age for both (Table 1).

The repeated-measures ANOVA on RT (congruent RT and incongruent RT) by class, sex, and AM reveal the significant main effects of the congruent/incongruent condition ($F_{1,1115} = 709.08$, $p < 0.001$, $\eta^2_p = 0.39$), class ($F_{7,1115} = 67.58$, $p < 0.001$, $\eta^2_p = 0.30$), and sex ($F_{1,1115} = 6.24$, $p = 0.013$, $\eta^2_p = 0.01$). The interaction between the congruent/incongruent condition and class is also significant ($F_{7,1115} = 11.09$, $p < 0.001$, $\eta^2_p = 0.07$). Post hoc analyses confirm that there is a flanker effect for all the class levels (all ps < 0.05) and that the RT in congruent conditions decreases with classes, with a more pronounced slope in the first five classes: class 1 differs from classes 3 and above, class 2 and 3 differ from classes 4 and above, class 4 differs from class 5, and class 5 and 6 differ from class 8. RT in the incongruent condition also decreases with the classes: classes 1, 2, and 3 differ from classes 4 and above, class 4 differs from all the subsequent classes, class 5 differs from classes 7 and 8, and classes 6 and 7 differ from class 8 (Table 1; Figure S2b). Sex difference shows that boys' responses are faster than those of girls, regardless of class or condition (1535.53 ms vs. 1587.42 ms). The variability is higher for RTs than for accuracy, and the ranges are between 0.14 and 0.19, and between 0.13 and 0.19 for congruent and incongruent RTs, respectively (Table 1).

*3.4. Updating: N-Back Task*

The ANOVA for the 1-back transformed scores (ranks) shows the significant main effect of class ($F_{7,1168} = 27.29$, $p < 0.001$, $\eta^2_p = 0.14$). Post hoc tests reveal that the scores in class differ significantly from classes 3 and above; the scores in classes 2 and 3 are lower than the scores in classes f4 and above; and the scores in class 4 are lower than the scores in classes 7 and 8 (all ps < 0.01). The scores in class 5, 6, 7 and 8 do not differ from each other (Table 1, Figure S3). There are no sex or AM effects, not even in interaction with each other, and the variability is lower than 10% for all the classes (CVs range 0.03–0.09, Table 1).

The ANOVA for the 2-back raw scores shows the significant main effect of class ($F_{7,1151} = 33.69$, $p < 0.001$, $\eta^2_p = 0.17$). Post hoc tests show that the scores in class 1, 2, and 3 differ significantly from class 4 and above; scores in class 4 are lower than scores in class 7 and 8; scores in class 5 are lower than scores in class 8 (all ps < 0.05). Scores in class 6, 7 and 8 do not differ from each other (Table 1). There are no sex or AM effects, not even in

interaction with each other, and the variability is lower than 10% for all classes (CVs range 0.06–0.08, Table 1; Figure S3).

### 3.5. Flexibility: Flanker Task

The repeated-measures ANOVA on accuracy (mixed congruent CR vs. mixed incongruent CR) by class, sex, and AM reveals the significant main effects of the congruent/incongruent condition ($F_{1,1066} = 1367.30$, $p < 0.001$, $\eta^2_p = 0.56$), class ($F_{7,1066} = 31.73$, $p < 0.001$, $\eta^2_p = 0.17$), and AM ($F_{1,1066} = 5.93$, $p = 0.015$, $\eta^2_p = 0.01$). The interaction between condition and class was significant ($F_{7,1066} = 15.64$, $p < 0.001$, $\eta^2_p = 0.09$). Post hoc analyses confirm that the accuracy is higher for the congruent than the incongruent trials at all class levels (all ps < 0.001). Congruent trials accuracy in the first four classes is lower than in the last ones; incongruent trials accuracy in the first classes differs from that of the intermediate classes, which in turn is lower than the accuracy in the last classes (Table 1; Figure S4a). The interaction between the condition and AM ($F_{1,1066} = 9.93$, $p = 0.002$, $\eta^2_p = 0.01$) is significant, which demonstrates that accuracy scores are lower in incongruent than congruent trials (all ps < 0.001) and that incongruent trials are less accurate in the remote condition ($p = 0.002$). The triple interaction condition X class X AM is also significant ($F_{7,1066} = 2.21$, $p = 0.031$, $\eta^2_p = 0.01$), but it is important to note that none of the scores in the in-person condition differ from the corresponding remote score. This interaction shows a significant congruent–incongruent effect for all classes in both assessment conditions. In the case of congruent trials in the in-person condition, there are the following significant differences between classes: 1 < 8; 3 < 5, 7–8; and 4 < 8. In the case of congruent trials in the remote condition, there are the following significant differences between classes: 1 < 7–8; 2 < 5–8; and 3–4 < 8. In the incongruent trials in-person, there are the following significant differences between classes: 2 < 7–8; 3 < 5–8; 4 < 7–8; and 5 < 8. In the incongruent trials in the remote condition, there are the following significant differences between classes: 1 < 4–8; 2 < 3–8; 3 < 5–8; and 4 < 6–8 (Table S2). The variability ranges between 0.04 and 0.13, and between 0.16 and 0.31 for congruent and incongruent correct responses, respectively, showing a decreasing trend with age for both (Table 1).

The repeated-measures ANOVA on RT (mixed congruent RT and mixed incongruent RT) by class, sex, and AM reveals the significant main effects of the congruent/incongruent condition ($F_{1,1077} = 1139.86$, $p < 0.001$, $\eta^2_p = 0.51$), class ($F_{7,1077} = 12.01$, $p < 0.001$, $\eta^2_p = 0.07$), and sex ($F_{1,1077} = 13.86$, $p < 0.001$, $\eta^2_p = 0.01$). The interaction between congruent/incongruent condition and class is also significant ($F_{7,1077} = 2.22$, $p = 0.031$, $\eta^2_p = 0.01$). Post hoc analyses confirm that the congruent trials are faster than the incongruent trials at all class levels (all ps < 0.001). The RTs in congruent condition decrease gradually with the classes, with significant differences between the first four classes and class eight. The RTs in the incongruent condition also decrease with classes: classes 1 and 2 are different from class 8, classes 3 and 4 are different from classes 5 and 6, respectively, and class 5 is different from class 8 (ps < 0.05; Table 1; Figure S4b). The sex main effect shows that boys' responses are faster than the responses of girls (1090.57 ms vs. 1144.12 ms). The variability ranges between 0.13 and 0.18, and between 0.12 and 0.17, for congruent and incongruent RTs, respectively (Table 1).

### 3.6. Daily Planning Test

The ANOVA by class, sex, and assessment modality for the DPT score shows a significant main effect of class ($F_{5,154} = 3.24$, $p = 0.008$, $\eta^2_p = 0.10$). Post hoc analyses show that the scores for class 3 differ from the scores for classes 4, 6, and above. No other differences were found between classes. The variability is around 10% for all classes (CVs range 0.07–0.12, Table 1; Figure S5).

### 3.7. Internal Consistency and Test Retest Reliability

All tasks show an acceptable to excellent internal consistency [36]. The Cronbach alpha is 0.93 and 0.68 for the Go CR and NoGo CR indices, respectively; 0.92 and 0.93 for

the congruent and incongruent trials of the single-rule flanker task, respectively; and 0.91 for the 1-back task and 0.75 for the 2-back task. 0.91 and 0.86 for congruent and incongruent trials of the mixed rules flanker task, respectively. The test–retest reliability is significant for all EF measures (all ps < 0.01; Table 2).

**Table 2.** Partial correlation between EF measures (control by age) and test–retest reliability.

|  |  | 1 | 2 | 3 | 4 | 5 | 6 | 7 | 8 | 9 |
|---|---|---|---|---|---|---|---|---|---|---|
| 1 | NoGo CR | — |  |  |  |  |  |  |  |  |
|  | *p* | — |  |  |  |  |  |  |  |  |
| 2 | Incongruent CR | 0.23 | — |  |  |  |  |  |  |  |
|  | *p* | <0.001 | — |  |  |  |  |  |  |  |
| 3 | Incongruent RT | 0.14 | −0.28 | — |  |  |  |  |  |  |
|  | *p* | <0.001 | <0.001 | — |  |  |  |  |  |  |
| 4 | Mixed Incongruent CR | 0.22 | 0.42 | −0.30 | — |  |  |  |  |  |
|  | *p* | <0.001 | <0.001 | <0.001 | — |  |  |  |  |  |
| 5 | Mixed Incongruent RT | 0.23 | 0.16 | 0.45 | −0.03 | — |  |  |  |  |
|  | *p* | <0.001 | <0.001 | <0.001 | 0.385 | — |  |  |  |  |
| 6 | 1-back CR | 0.27 | 0.30 | −0.11 | 0.27 | 0.06 | — |  |  |  |
|  | *p* | <0.001 | <0.001 | <0.001 | <0.001 | 0.060 | — |  |  |  |
| 7 | 2-back CR | 0.28 | 0.33 | −0.16 | 0.32 | 0.04 | 0.55 | — |  |  |
|  | *p* | <0.001 | <0.001 | <0.001 | <0.001 | 0.154 | <0.001 | — |  |  |
| 8 | DPT | 0.16 | 0.28 | −0.20 | 0.17 | 0.07 | 0.17 | 0.17 | — |  |
|  | *p* | 0.039 | <0.001 | 0.008 | 0.032 | 0.343 | 0.031 | 0.031 | — |  |
| 9 | QUFE Parents | 0.12 | 0.13 | −0.04 | 0.13 | −0.03 | 0.12 | 0.13 | 0.08 | — |
|  | *p* | 0.003 | 0.003 | 0.319 | 0.003 | 0.534 | 0.006 | 0.002 | 0.332 | — |
| 10 | QUFE Teachers | 0.20 | 0.24 | 0.00 | 0.18 | 0.23 | 0.19 | 0.27 | 0.33 | 0.34 |
|  | *p* | <0.001 | <0.001 | 0.919 | <0.001 | <0.001 | <0.001 | <0.001 | <0.001 | <0.001 |
|  | Test retest reliability | 0.43 | 0.38 | 0.64 | 0.52 | 0.67 | 0.42 | 0.61 | - | - |
|  | *p* | 0.002 | 0.006 | <0.001 | <0.001 | <0.001 | 0.003 | <0.001 |  |  |

*3.8. Partial Correlations*

Partial correlation analyses were executed between the EF indicator of each task, i.e., NoGo CR, incongruent CR, and RT in the single-rule flanker task, 1- and 2-back CR, DPT, and total scores of the QUFE questionnaire (parent and teachers; Table 2). All direct measures are significantly associated with each other, except for RTs of mixed incongruent trials. The indirect measures, i.e., QUFE total scores are also correlated with the accuracy measures of all tasks, except for the absence of correlation between the parent report score and the planning score.

## 4. Discussion

The tele-assessment of cognitive functioning in children is a debated question, mainly because most of the existing tests are normed and standardized for in-person administration. Considering verbal ability or memory, the validity of remote administration is quite widespread and recognized, as the tests used are based on oral presentation [11,37]. On the contrary, there is less agreement concerning other cognitive processes that are typically assessed with paper and pencil tests or object manipulations, as in the case of EF [3]. Despite some evidence existing about the possibility of comparing the two forms of assessment also with EF tasks [11], there is a need to establish standardized procedures and psychometric properties of instruments for this assessment [5]. Developing and standardizing tests for remote administration can help in reducing the critical issues that have so far characterized tele-assessment.

In this vein, TeleFE is a web platform for both remote and in-person administration. The aim of the present study is to explore the effects of assessment modality (remote vs. in-person) on EF measures in school-age children and adolescents. In addition, class and sex differences in task performances were investigated. The results show that TeleFE is a valid instrument to assess EFs in children. First, negligible differences between in-

person and remote evaluation have been found in the TeleFE tasks. The difference between assessment modalities emerges in the interaction with congruent/incongruent conditions in the flanker task considering the flexibility accuracy, which confirms the flanker effect for both the assessment conditions and reveals a lower accuracy in the incongruent trials that are remotely assessed. The results also suggest that the differences between classes in the congruent and incongruent trials may be modality dependent, while no difference between the correspondent remote/in-person comparisons within each class and condition emerges. This result is very relevant, given the distrust with which the remote assessment of skills other than language and memory is viewed with [4]. It shows that the two assessment modalities are equally valid when higher-order cognitive abilities, as those implicated in the self-regulation of thoughts and behavior, are considered. Second, the results on class-related differences in task performances show the instrument's ability to capture developmental differences, thus confirming its usefulness for clinical practice. Such results are in line with the existing literature, showing a continuous maturation of EF through childhood, with a period of accelerated development of EF up to 8 years, followed by a more gradual development in the subsequent years [15,38]. Concerning sex-related differences, the results show that males are more impulsive, showing a lower accuracy score in the NoGo CR. In addition, males show faster responses in both Go/NoGo and flanker tasks in comparison to females. Such results are not unexpected, as higher inhibitory abilities in females are well recognized [39–42]. As repeatedly replicated in the literature, the correlations between cognitive EF measures are low to moderate [23]. The absence of association between RTs in the shifting task and the other EFs measures, except for inhibitory measures, suggests being cautious in using this indicator in the assessment of EF because fast or slow RTs can be influenced by individual differences and do not necessarily correspond to a high or low level of performance in shifting. The literature shows that RTs as measures of shifting are often uncorrelated with working memory measures, therefore suggesting the use of combined accuracy-RT measures. Another non-negligible aspect concerns the delayed differentiation of dimensions such as shifting [20] that may require a thorough interpretation of the results, especially until middle childhood. Indirect and direct measures of EFs are significantly associated with each other, even if all coefficients are small, indicating an expected low association between the considered measures [23]. Consistently to the literature, the interrater correlation is moderate [23]. Considering that parent reports appear to be poorly correlated with the direct measures, especially for RTs or planning measures, we may hypothesize that parents may focus on different aspects of their child's performance and are less sensitive to characteristics such as timing. Thus, different raters' reports reflect non-overlapping information in EFs-related behaviors; in addition, they may be differently affected by the raters' biases, such as halo and leniency bias. In other words, the teacher reports provide information on EFs-related behaviors more strictly associated with the cognitive measures of EFs.

Even though our results support the feasibility of the remote cognitive assessment, some considerations had to be thought about when choosing the tele-assessment. First, in tele-assessments, interactions are technology-mediated, with possible consequences on the child's behavior and compliance during testing. A further variable is the family and the child's access to technology. For instance, a socio-economically disadvantaged family may not have the necessary devices or be familiar enough with them [11]. In addition, children with impaired motor functions may not be able to use the keyboard as required by the tele-assessment tool. This is an important limitation of remote assessment and telehealth in general; however, the positive impact of tele-assessment is still considerable. For example, it can be an important resource for public services. The possibility to use tele-assessment with at least some of the service afferents reduces the number of in-person accesses, which can then be devoted to a greater extent of those cases that do not allow the use of tele-assessment because of their SES or other conditions. In general, although there is no doubt that tele-assessment is not appropriate in all cases, having a tool to choose which mode of assessment to adopt is an important breakthrough for clinicians. Another

important factor to consider concerns the age of the children. In our study, the percentage of children not reaching the threshold scores set to consider the assessment reliable is higher in younger children in the remote modality. This is evident in the single-rule flanker task. Of the children removed from the analysis, 45% were in the first three classes. Such results indicate to be careful using tele-assessment with young children. Even if TeleFE allows the administration of tasks, both in-person and remotely from the age of 6, the clinician must prefer the in-person evaluation for children younger than 6–7 years when possible. This is not totally unexpected, as younger children are most in need of adult support during a tele-assessment.

In addition, the results may indicate that, while during an in-person evaluation the clinician can notice the child's difficulties and can repeat the test, if necessary, this is more difficult in the case of remote evaluation. Therefore, the clinician needs to observe the child's behavior during the tests and consider any aspect that could influence his or her performance. As for the flanker task, a helpful strategy to reduce the risk of not interpretable or inaccurate results could be to put stickers on the keyboard keys needed to perform the task. In fact, while the Go/NoGo and N-back tasks require only to press the spacebar, the flanker task requires the child to press two different keys depending on the stimuli, and it would be difficult for younger children to maintain the correct position. Finally, as for the in-person assessment, it is important to combine the data obtained from different tests, questionnaires, and interviews that are also collected in-person. According to previous studies [43], it is preferable to use remote tools as a support for a broader assessment that includes an in-person assessment, and it is particularly helpful in screening phases [44].

Finally, some limitations should be considered. First, the sample of children in the first and second classes is smaller than that of the other classes, and a higher percentage of them are evaluated remotely. In the future, results should be confirmed with an increased sample of children assessed in-person. Second, reliability measures for the DPT could not be provided. It will be necessary to obtain these measures in the future. In addition, only typically developing children were included in the sample. Even though previous studies demonstrate the validity of this assessment modality also for children with developmental disorders [4], future studies should investigate the clinical relevance of the tool both in terms of its usability and feasibility with this population, and in terms of the discriminative validity of the tool.

## 5. Conclusions

In conclusion, the present results support and advance the evidence on the validity of the tele-assessment of the cognitive functions of children, also for higher-order abilities such as EF. They also offer a series of helpful directions to guide the clinicians in using tele-assessment and interpreting the results. In line with the recent APA guidance on psychological tele-assessment [10], it is important for the psychologist to know the limits of tele-assessment and to consider if this approach is appropriate given the referral question, evidence, client characteristics, and his/her access to adequate assessment space and conditions (i.e., a quiet room, a safe, and a functional connection). In addition, clinicians must be aware of the possible limitations of remote evaluation and think through the quality of the data collected, as aspects such as low task comprehension or technical issues can affect performances and are not easily detected in cases of remote evaluation. Overall, the study shows that tele-assessment is a valuable tool that brings many benefits to the cognitive assessment of children, but it cannot replace face-to-face assessment, which still appears more appropriate in certain circumstances.

**Supplementary Materials:** The following are available online at https://www.mdpi.com/article/10.3390/app13031728/s1. Figure S1: means and standard deviations for each measure of the Go/NoGo task by class, Figure S2: means and standard deviations for each measure of the single-rule flanker task by class, Figure S3: means and standard deviations for each measure of the N-back task by class, Figure S4: means and standard deviations for each measure of the mixed-rule flanker task by class, Figure S5: means and standard deviations of the percentage of accuracy on DPT by class,

Table S1: estimated marginal means of accuracy condition (congruent–incongruent) * class* AM for blocks 1 and 2 in the flanker task, and Table S2: estimated marginal means of accuracy condition (congruent–incongruent) * class* AM for block 3 in flanker task.

**Author Contributions:** Conceptualization, C.B., G.M.M., C.P., C.R. (Carlotta Rivella), C.R. (Costanza Ruffini), L.T., M.C.U. and P.V.; methodology, G.M.M., C.P., M.C.U. and P.V.; software, A.F.; formal analysis, C.R. (Carlotta Rivella) and M.C.U.; investigation, A.C., A.M., C.R. (Carlotta Rivella), C.R. (Costanza Ruffini) and L.T.; resources, A.C., A.F., G.M.M., A.M., C.P., C.R. (Carlotta Rivella), C.R. (Costanza Ruffini), L.T., M.C.U. and P.V.; data curation, C.R. (Carlotta Rivella), C.R. (Costanza Ruffini) and M.C.U.; writing—original draft preparation, C.R. (Carlotta Rivella) and M.C.U.; writing—review and editing, G.M.M., C.P., C.R. (Carlotta Ruffini), L.T. and P.V.; visualization, C.R. (Carlotta Rivella) and M.C.U.; supervision, G.M.M., C.P., M.C.U. and P.V.; project administration, G.M.M., C.P., M.C.U. and P.V.; funding acquisition, G.M.M., C.P. and P.V. All authors have read and agreed to the published version of the manuscript.

**Funding:** This research was founded by the Ministry of Education University and Research, grant number 1760961.

**Institutional Review Board Statement:** The study was conducted according to the guidelines of the Declaration of Helsinki and was approved by the Ethics Committee of the University of Genoa (protocol code 2022/16, 17/02/2022).

**Informed Consent Statement:** Informed consent was obtained from all parents of the subjects involved in the study.

**Data Availability Statement:** The data presented in this study are available on request from the corresponding authors. The data are not publicly available due to privacy restrictions.

**Conflicts of Interest:** The authors declare no conflict of interest.

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
