# Peer review of "TeleFE: A New Tool for the Tele-Assessment of Executive Functions in Children"

_applsci, doi:10.3390/app13031728_

Round 1

Reviewer 1 Report

1.Please give the specific P values in Table2. 2.In the abstract section, please give the specific results and corresponding P values. 3.For all other parts, the specific P value should be given, not the range.

Author Response

Thank you for the suggestion. We added the exact p values in table 2 and in the abstract section. In the text, we already used the standard notation for p values. We report the exact p value (or the standard notation p<.001 for values smaller than .000) for all the results except for the post hoc analysis. This is to make the reading smoother and not to make the text and tables too dense. However, if you find it useful and necessary, we can provide tables with all post-hoc comparisons to be included in supplementary materials.

Reviewer 2 Report

In this manuscript, the authors compared the validity of the TeleFE measure when administered remotely and in-person. With the exception of the Flanker task, most tasks worked equally well in a remote and in-person setting. As expected, there were age related improvements in TeleFE scores from grades 1-8. This study is an important contribution as the field needs reliable and valid remote assessments of EF; the manuscript was also very well written and easy to follow. However, it could be improved by including more detail in the Method section and expanding the Discussion section. Please see below for more details on these comments, as well as more minor suggestions that I hope will be helpful for the authors as they refine the paper.

Introduction

-       In section 1.1 the authors do a nice job motivating the benefits of remote assessment, but it does seem a little one sided. I would suggest that they include a more complete overview of tele-assessments that note some of the limitations. For example, families who do not have access to technology or do not have the right kind of technology (e.g., own a smartphone, but not a computer) and how that has been (or might be) addressed.

-       Given the large age range of the participants in the study, it seems important to also include the developmental challenges of measuring EF. In section 1.2 there is a great discussion of the task impurity problem, but no mention of how the structure of EF changes over development (e.g., from more unitary to more distinct skills).

Method

-       There are some key details missing from the Method section, including the race/ethnicity of the children, any indication of socioeconomic background of the families, and if/how the families were compensated for participating.

-       For the procedure, were there any explicit instructions for the adults who were helping the children with the remote assessment (e.g., where to sit, to remain quiet, etc.)?

-       How long was a typical study session?

-       The description of each task is clear and concise. Were there any practice trials included? And, if so, did the task proceed if children failed the practice trials?

-       The authors note that they excluded children who had less than 60% accuracy. What was the motivation for using this benchmark? Are there previous studies the authors can include to justify this decision? Given the nature of EF and that children with weaker EF skills tend to perseverate, it is common to see below chance performance on these types of tasks.

Results

-       It would be helpful to also see the internal consistencies for the parent and teacher questionnaires.

Discussion

-       It seems to me that a major limitation is that those who were excluded from the analyses tended to be younger children who have less well-developed EF skills. The authors should discuss this issue in more detail. How can we be sure that these measures are valid for all children if those with weaker EF skills were not included?

-       I would also like to see a more in-depth discussion of the youngest children struggling the most with the remote assessment of the Flanker. Surprisingly, grade 1 children performed better on the Incongruent trials than the Congruent trials in the tele-assessment. The authors briefly note that there may have been insufficient support during the remote assessment, but what might be specific ways to improve task administration moving forward?

-       In general, the Discussion section is missing commentary on the performance-based vs. report-based measures. One particularly interesting finding is that the teacher report tended to be more highly correlated with the performance-based measures than the parent report (this is most obvious with the Mixed Incongruent RT and 2-back CR). Do the authors have any thoughts on why this might be?

-       There is also no discussion about the correlations among the EF tasks. It seems to me that the authors could highlight which measures seems to be working best. Based on these correlations, the Mixed Incongruent RT scores may not be particularly meaningful and, in general, the Flanker task is functioning more poorly than the other tasks.

Author Response

Thank you for your insightful comments.  Attached the file with responses.

Reviewer 3 Report

This article deals with the problem of cognitive tele-assessment, specially  in children. Recent literature demonstrates the validity of cognitive tele-assessment in youth in a vast vary of cognitive domains. The current find out about aimed to current TeleFE, a new device for the tele-assess-19 types of EF in adolescents aged 6-13. TeleFE consists of a net platform such as 4 tasks based totally on 20  neuropsychological paradigms to consider inhibition, working memory, cognitive flexibility, and daily planning. Results highlight the interchangeability of the two evaluation modalities (remote vs in person) at each stage of development. In addition, consequences display the sensitivity of TeleFE to age-related variations in the improvement of EF. 

The article is well written, all the sections are well structured and provides all the relevant and necessary information to understand the article.

I leave a couple of comments to clarify some doubts:

Line 141: the authors says "stimuli size was calibrated before the beginning 141 of the assessment by matching a one Euro coin on its image on the screen", but they do not clarify if this calibration were also carry out in remote mode. And if yes, Who did this calibration? how was it done?

Line 160: in Go/NoGo task and Flanker task tests the authors do not clarify what happened if a child had an erroneous or unexpected behavior. For example, a key pressed with no enough strength, or the child change accidentally the possition of the finger in the key board. So, the child behaviour is correct, but the program do not record the answer as right, or the RT is higher than expected. 

Can you clarify this?

Tnak you very much.

  •  

Author Response

This article deals with the problem of cognitive tele-assessment, specially in children. Recent literature demonstrates the validity of cognitive tele-assessment in youth in a vast vary of cognitive domains. The current find out about aimed to current TeleFE, a new device for the tele-assess-19 types of EF in adolescents aged 6-13. TeleFE consists of a net platform such as 4 tasks based totally on 20  neuropsychological paradigms to consider inhibition, working memory, cognitive flexibility, and daily planning. Results highlight the interchangeability of the two evaluation modalities (remote vs in person) at each stage of development. In addition, consequences display the sensitivity of TeleFE to age-related variations in the improvement of EF. 

The article is well written, all the sections are well structured and provides all the relevant and necessary information to understand the article.

I leave a couple of comments to clarify some doubts:

Line 141: the authors says "stimuli size was calibrated before the beginning of the assessment by matching a one Euro coin on its image on the screen", but they do not clarify if this calibration were also carry out in remote mode. And if yes, Who did this calibration? how was it done?

Thank you for your comments. We modified as follow: regardless of the modality used, before the beginning of the assessment, the adult assisting the children was required to do the calibration of the stimuli. Stimuli size was calibrated by matching a one Euro coin on its image on the screen.

Line 160: in Go/NoGo task and Flanker task tests the authors do not clarify what happened if a child had an erroneous or unexpected behavior. For example, a key pressed with no enough strength, or the child change accidentally the possition of the finger in the key board. So, the child behaviour is correct, but the program do not record the answer as right, or the RT is higher than expected. 

In order to reduce the risk of erroneous behavior that could distort the results, such as a key pressed with not enough strength or an incorrect button pressed, each task includes a training session. In the first phase of the training session, the buttons the children have to press appear on the screen. Then, a trial of the task appears and the child is asked to press the correct button. In case of remote administration, the examinator sees on the screen if the child’s response is correct or not. In case of an incorrect response, the examinator can verify if it is due to a misunderstanding of the task or to an erroneous behavior. The training session ends when the examinator is sure that the child has understood the task and the right behaviors during the task. In addition, the assessment takes place in the presence of an adult who can report to the examinator any unexpected events without interfere with the evaluation.